# Use of Polar Heliostats to Improve Levels of Natural Lighting inside Buildings with Little Access to Sunlight

**DOI:** 10.3390/s22165996

**Published:** 2022-08-11

**Authors:** Luis Manuel Fernández-Ahumada, Maximiliano Osuna-Mérida, Jesús López-Sánchez, Francisco Javier Gómez-Uceda, Rafael López-Luque, Marta Varo-Martínez

**Affiliations:** 1Department of Electrical Engineering and Automatics, University of Córdoba, 14071 Córdoba, Spain; 2Research Group Physics for Renewable Energies, University of Córdoba, 14071 Córdoba, Spain; 3Department of Applied Physics, Radiology and Physical Medicine, University of Córdoba, 14071 Córdoba, Spain

**Keywords:** natural lighting, polar heliostat, energy efficiency, sustainability, communications, automated solution

## Abstract

The growing need to increase environmental and energy sustainability in buildings (housing, offices, warehouses, etc.) requires the use of solar radiation as a renewable source of energy that can help to lower carbon footprint, making buildings more efficient and thereby contributing to a more sustainable planet, while enhancing the health and wellbeing of its occupants. One of the technologies deployed in the use of solar energy in buildings is heliostats. In this context, this paper presents an analysis of the performance of a heliostat illuminator to improve illumination in a classroom at the Campus of Rabanales of the University of Cordoba (Spain). A design of a system in charge of monitoring and measuring daylighting variables using Arduino hardware technology and free software is shown. This equipment develops the communications, programming and collection of lighting data. In parallel, installation of an artificial lighting system complementary to the natural lighting system is implemented. Finally, an analysis of the impact of the proposed solution on the improvement of energy efficiency is presented. Specifically, it is estimated that up to 64% of savings in artificial lighting can be achieved in spaces with heliostatic illuminators compared to those without them.

## 1. Introduction

Artificial lighting is one of the most important energy loads in buildings [1,2,3,4,5], and this load is expected to increase due to population growth and the search for improved quality of life [6,7,8,9]. In addition, the lighting systems chosen in the design of buildings have an impact on their thermal conditions, which, in turn, influences energy consumption and the comfort of the occupants [10].

In this regard, it is important to point out that the use of sunlight as a source of natural lighting has important advantages over the use of other lighting systems. In the first place, natural lighting, based on the use of solar resources, leads to savings in electrical energy consumption compared to artificial lighting systems and improves the energy efficiency of buildings [8,11,12]. This energy saving not only has a lower economic cost but also an environmental one since it does not require energy sources with carbon dioxide emissions. Furthermore, natural lighting improves the wellbeing of the occupants [5,8,10,13,14]. On the contrary, a lack of natural light alters the light rhythms of people [15,16], negatively affects the sense of orientation and time [17] and can cause claustrophobia, anxiety and depression [17,18]. For all these reasons, it is important that policies to improve the energy efficiency of buildings include optimisation of natural lighting systems versus artificial ones as one of the central aspects.

There are different types of natural lighting systems: lateral, ceiling or combined. The choice between them depends on the climate, the architectural environment or the use of the building, among other factors [1]. The most frequent natural lighting systems are the lateral ones. However, it is difficult to achieve good natural lighting conditions in large and deep spaces, such as classrooms and offices, if only the side windows of the building are used [11,12]. In this regard, Stetsky and Shchelokova [19] state that ceiling natural lighting systems, such as skylights, sheds or monitors, are more effective than lateral systems. When only lateral natural lighting systems are used, the points furthest from the windows may not receive adequate lighting levels [11]. In these cases, to make use of natural lighting, installing sun pipes, fibre or light shelves can be resorted to. Specifically, Mohapatra et al. [11] experimentally simulated and measured in a scale prototype the levels of natural lighting that are achieved in a large and deep space when light shelves are used as a natural lighting element complementary to side windows, finding that they are effective. Similarly, simulations of the behaviour of light pipes as a natural lighting element show that they are effective in achieving adequate and homogeneous natural lighting and that they lead to significant energy and economic savings [20].

Thus, these passive natural lighting systems take advantage of the solar radiation that falls on them and redirects it to points where it does not reach naturally. In contrast, Whang et al. [9] proposed an active illuminator system that, in addition to redirecting sunlight towards the point of interest, incorporates solar tracking techniques that allow us to increase the levels of solar radiation used as a basis for natural lighting. One of the devices that allow this objective to be achieved is the heliostat. A heliostat is a system of mirrors that follows the movement of the sun while redirecting reflected solar radiation in a fixed direction. Although their best-known use is as a concentrator in solar thermal power plants, heliostats can also be used as natural lighting systems in buildings [21]. Therefore, in recent years there has been a growing trend in the use of heliostatic illuminators. One example is Torres-Roldan et al. [22], who developed a simple, low-cost, single-axis polar heliostat prototype applicable to the design of daylighting systems in buildings. In addition, although it is a polar system, it was verified that it presents a high level of pointing both when it is aligned with the axis of the Earth [22] and when, due to architectural needs, it is necessary to orient it in another direction [21]. Likewise, Whang et al. [9] proposed an active illuminator consisting of a heliostat and a mirror system. Specifically, thanks to the solar tracking of the heliostat, the levels of solar radiation are increased, and the modular system of mirrors provides uniform lighting on the plane.

Furthermore, visual comfort is related to natural light that helps humans access visual information without disturbing their visual senses [12,23], as both the lack and excess of light generate eye fatigue [23]. Visual comfort is especially important in teaching spaces since inadequate lighting levels can negatively affect the learning process [23]. Various studies confirm that adequate levels of natural lighting in teaching spaces improve the level of student satisfaction, having an impact on improving the learning process [10]. This is motivated, to a certain extent, by the fact that natural lighting affects the biological processes of the human body from a hormonal point of view. As a consequence, exposure to natural light contributes to the regulation of day–night cycles, favours students’ rest and improves their ability to concentrate [24]. For this reason, it is important to ensure that architects who design buildings for educational use, whether newly built or refurbished, promote the use of natural lighting as the main source of lighting in classrooms. Kwon and Lee [10] proposed a methodology for the design of educational buildings that, through the combination of simulation software and passive design methods, seek to optimise the levels of natural lighting in classrooms with different orientations. Likewise, Natalia and Suharjanto [23] simulated the lighting levels inside a classroom for different types of windows and their opening area, finding that the capture of natural lighting is optimised for a window opening area of 60% of the wall area.

The use of electronic control systems in lighting has been common for years, but not in applications that combine heliostats [25]. Monitoring is considered to be a basic task, together with the placement of sensors, especially in the school environment [26]. The development of non-proprietary control systems has made these tools available, and the concept of Open Science has been facilitated in this field of interior lighting [27]. Through electronic monitoring and control, a useful tool has been provided for the evaluation and validation of saving strategies in different buildings and rooms [28].

In accordance with the above, this paper presents a study on the possible application of heliostats as devices to improve natural lighting in educational spaces. To do so, a scale model of a real classroom was reproduced, and heliostatic illuminators were implemented in it, analysing the improvements achieved in the levels of natural lighting and in the energy efficiency of the space. The present work advances the use of scale models that can be considered as physical scale twins to approximate solutions to real daylighting situations that would be more complex to study at full scale [29]. Although attempts have been made to minimize possible scale effects as well as differences in optical properties between model materials and real space, the authors are aware of the need to continue to evaluate and correct for such effects. It is expected that the final phases of the line of research presented here will involve obtaining the aforementioned physical scale model or physical twin on which the behaviour of the real room can be tested.

After this introduction, in which a review of the state of science on natural lighting is presented, Section 2 explains the methodology followed for the construction of the scale model of both the classroom and the heliostats, the design of the electronic device developed to monitor the illuminance levels inside the classroom and the strategy followed to take experimental measurements in conditions similar to those of the real classroom that is reproduced in the scale model. Section 3 presents useful initial results based on average classroom illumination. Section 4 presents the main conclusions already valid from the initial phase concerning the development of the scaled twin as well as future work.

## 2. Methodology

To analyse the possible use of heliostats as illuminators in spaces with little or no access to solar resources, a scale study of a real situation was designed. Specifically, the case of the Rabanales University Campus of the University of Córdoba (Spain) was analysed. Despite the high number of hours of sunshine recorded in Córdoba (latitude 37.85° N, longitude 4.78° W), several buildings and teaching spaces with zero or deficient access to solar resources were identified on this university campus, which has a negative impact on its efficiency and sustainability. An example of this is the Leonardo da Vinci building, in which most of its classrooms lack windows with direct exposure to the sun so the levels of natural lighting inside are low and insufficient for teaching (Figure 1).

For this reason, it is necessary to resort to artificial lighting throughout the interval of use which, except for weekends and non-teaching days, can reach 11 h of daily use from Monday to Friday from September to May. Therefore, this continuous need for artificial lighting contributes to increasing electricity consumption on campus which, according to data from the Environmental Protection Service of the University of Córdoba, in 2020, despite the closure of facilities during confinement motivated by COVID-19, reached 1951.5 kWh/person compared to the 837 kWh/person that is consumed globally throughout the institution [30]. In accordance with this, it is advisable to implement technological measures that improve the levels of natural lighting in the classrooms and contribute to reducing electricity consumption and improving the energy efficiency and sustainability of the campus.

In this study, a scale analysis of a system of heliostatic illuminators in the Leonardo da Vinci building was performed. As Figure 2 and Figure 3 show, the roof of the building is shaped like saw teeth. The vertical planes of the glazed saw teeth are oriented towards the North, while the oblique planes are oriented towards the South with an inclination of 25°. The solar radiation that penetrates the building through the vertical glazed planes of the roof falls on the upper face of the false ceiling of the classrooms, in which some plaster plates have been replaced by translucent plates (Figure 1). However, this measure is insufficient and, as can be seen in Figure 1, lighting levels, even on clear days, are very poor, making it necessary, as mentioned above, to resort to the use of artificial lighting systems.

Given these circumstances, it is proposed to install heliostatic illuminators on the saw teeth of the building roof that redirect solar radiation towards the interior of the classroom, as shown in Figure 4. This paper analyses the implementation of this proposal by simulating the behaviour of the system to scale.

### 2.1. Proposed Single-Axis Polar Heliostat as Illuminator

To develop this natural lighting system, it is proposed to use the single axis simple polar heliostat developed by Torres-Roldan et al. [22] that allows redirecting the sun’s rays towards a direction parallel to the axis of rotation of the Earth.

To do this, the heliostat (Figure 5) is made up of a deformable polygon A’DCBAA’ and a primary mirror supported on the DC side of this polygon in such a way that the planes of the mirror and the polygon are mutually perpendicular. By means of an Arduino board and mathematical models based on the laws of reflection and Earth–Sun movement, the motor responsible for the rotation of the A–A’ axis is controlled. When this axis rotates, the screw located at vertex B of the polygon is screwed around the axis, producing the deformation of the polygon and consequently the movement of the primary mirror. The rays are reflected along the polar axis direction through the secondary mirror that redirects rays towards the desired direction.

In accordance with the above, the proposed heliostat prototype presents an important advantage and innovation compared to commercial heliostats since, unlike the latter, it requires a single axis for solar tracking. This entails the simplification of the mobile mechanisms and, consequently, a significant reduction in costs without affecting the simplification in the focusing precision of the device, which is estimated to be around thousandths of a radian [21,22,31].

### 2.2. Scale Model of the Case Study

In order to study the effect on natural lighting of the installation of a polar heliostat system based on the prototype proposed by Torres-Roldan et al. [22], two scale models of a real classroom of the Leonardo da Vinci building of the Rabanales University Campus of the University of Córdoba (Spain) were developed. A system of heliostatic illuminators, also to scale, was implemented on one of these scale models, while on the other model, no action was taken. Both models were exposed to identical circumstances of solar incidence, and the lighting levels inside were monitored. The comparative study of these levels of natural lighting inside the prototypes allowed us to evaluate the effects of the lighting system based on heliostats.

For the construction of the models, a 1:15 scale was chosen so that their final dimensions were 100 cm × 72 cm × 60 cm (Figure 6a). Likewise, to provide greater realism and simulate the reflectance levels of the materials found in the real classroom, the interior was also reproduced in the model with the greatest possible precision (Figure 6b). To do this, the student desks and the teacher’s platform and table were recreated (using 18 cm × 5.5 cm × 2.5 cm wooden slats). Finally, stickers obtained from photographs of the real classroom treated with the CamScanner software were placed on the internal surfaces of the model. As can be seen in Figure 7, the level of similarity achieved is very high.

Once the models of the classrooms were built, the scale model of the heliostatic illuminators based on the prototype of Torres-Roldán et al. [22] was developed for the construction of the heliostat scale model (Figure 8a), and some constructive modifications were introduced, as shown in Figure 8b. The main change consists of the use of a rotating axis housed inside a threaded tube. In Figure 8b, the axis is shown in green, and the threaded tube is sectioned. Thus, the rotating motor rotates the set of articulated arms whose end is in the nut (yellow) threaded on the tube. As the tube remains fixed, the rotation of the structure causes the nut to ascend (or descend) and the plane of the mirror to acquire a variable angle relative to the axis of rotation. This mechanism allows the primary mirror to rotate redirecting rays along the polar axis direction. Both the Torres heliostat and the one modified in this work are polar, i.e., their reflected rays are always parallel to the Earth’s axis, so they require secondary mirrors (Figure 8c) to redirect the radiation in any direction (with a tuned configuration preventing the reflection in the primary mirror).

As for the classroom models, the chosen scale was 1:15, resulting in the primary reflectors of each heliostat measuring 5 cm × 5 cm. For the construction of these scale models, a stepper direct current motor with a threaded rod with a pitch of 1 mm was used that acts as the axis of rotation of the deformable polygon, metal rods, constant velocity joints, methacrylate and a specular surface (Figure 8a). The movement control of the primary reflector of the heliostat was carried out by means of a 28BYJ-48 unipolar stepper motor, an Arduino MEGA 2560 compatible board and a L298 power driver that was responsible for providing adequate power to the stepper motor.

The scale heliostats were installed on the roof of one of the two classroom models developed, leaving the other free of action. Since they are polar heliostats, their axis of rotation must have an inclination equal to the latitude of the place, in this case, 37°. For this reason, given that the inclined planes of the saw teeth on the roof of the building have an inclination of 31° with respect to the horizontal, it was necessary to install a wedge-shaped supplement that increases this inclination to 37° latitude in Córdoba. Figure 9 show the location plan of the heliostats in the classroom and the location of the heliostats in the model. With this spatial distribution, we attempted to produce a greater impact on the first rows of desks in which the students are concentrated during teaching activities.

### 2.3. Illuminance Level Monitoring System

Furthermore, an identical monitoring system was developed and implemented in both scale models to record the levels of natural lighting inside. Each of the systems consists of a set of 9 TSL 2561 illuminance sensors, an Arduino MEGA 2560 board based on the ATMEGA2560 microcontroller, a real-time clock RTC DS1302 and an SD card for data storage. The 9 TSL 2561 sensors were arranged in the classroom according to the scheme in Figure 10. The fact that more sensors were arranged on the right side of the classroom is due to the fact that, as can be seen in Figure 6, the blackboard is displaced towards that area; therefore, it was assumed that the preference of the students when choosing a seat would be greater for the column of desks on the right than on the left.

### 2.4. Artificial Lighting System

In accordance with the IES Lighting Book [32], the recommended lighting level for teaching spaces is 300 lx. However, natural lighting is variable throughout the day and year and depends on different weather factors, so it is not possible to guarantee this level of lighting coming only from the natural source. For this reason, it is advisable to install an artificial lighting system that complements the natural lighting received inside each model. In this regard, it is expected that the levels of natural lighting will be higher in the case of the classroom with heliostatic illuminators than in the classroom that does not have them, so the artificial lighting support needs will also be different in each case.

Therefore, an artificial lighting system was implemented that, through automatic regulation, complements and combines with the natural lighting system to achieve optimal lighting levels. This artificial lighting system is based on the use of LED technology, whose arrangement in the model is shown in Figure 11 by blue stripes.

The regulated lighting system that is controlled from the Arduino MEGA 2560 board through PWM modulation is made up of a MOSFET IRF530 transistor, an N2222 transistor and two resistors of 1 kΩ and 10 kΩ, respectively (Figure 12).

### 2.5. Automation System

According to what has been said, the Arduino MEGA 2560 board oversees managing and controlling the monitoring system, the regulated artificial lighting system and the movement of the stepper motors in charge of controlling the movement of the heliostats (Figure 13). The control strategy consists of a continuous reading of the illuminance through the sensors. Depending on whether the minimum value established for this type of room is reached, the heliostats (which increase the natural illumination) are activated, or the system is left with sunlight. If the minimum illuminance is not achieved with natural light (heliostats plus room illumination), artificial lighting is used. Figure 14 show the connection of the Arduino MEGA 2560 board.

### 2.6. Data Register

Once the design and construction phase of the scale models and the implementation of the natural and artificial lighting systems and their respective control and monitoring systems were completed, the experimental data collection phase of the lighting levels in the interior of each of the two scale models (with and without heliostatic lighting system) was carried out. To do this, both models were placed outdoors in an area where they were not shaded by any element and with the same orientation as the replicated real classroom in order to simulate the natural lighting conditions in the classroom (Figure 15a). Figure 15b show the ray’s trajectory over the system. Simultaneously, the experimental values of global and diffuse solar irradiance were recorded. Illuminance data were recorded every 10 min from February 2022 through June 2022.

## 3. Results

From the recorded data, comparative analysis was carried out, both qualitatively and quantitatively, of the lighting levels inside both models.

From a qualitative point of view, Figure 16 show the photographs of the interior of both models captured by webcam on 3 July 2022 at 12:19 p.m. under the same conditions of solar incidence. It was observed that the levels of natural lighting inside the scale model of the classroom with heliostats (without artificial lighting supplement) were higher than in the model without heliostats. Therefore, it was confirmed that the artificial lighting needs are higher in the classroom without heliostatic illuminators, which will result in higher consumption and lower energy efficiency.

For quantitative analysis, the lighting levels recorded by each luxmeter inside both models were compared (Figure 17).

It was observed that the average lighting levels recorded by all the sensors were higher in the classroom with heliostatic illuminators than in the classroom without them, with increases (red line) that varied between 70.4% (sensor 9) and 242.58% (sensor 4). Similarly, the maximum lighting levels were also higher for all sensors in the case of the classroom with heliostats.

Moreover, the maximum lighting levels in the classroom with heliostatic illuminators were always higher than 300 lx (except for the case of sensor 8, with 294 lx), and the average levels were close to or exceed 200 lx in all cases. However, in the case of the classroom without heliostats, the average levels did not reach 100 lx in any case, and the maximum levels were less than 160 lx for all sensors. In fact, except in the back of the room, the maximum levels of the room without heliostats were lower than the average levels of the room with heliostats. This implies a greater need for artificial lighting in the case of the classroom without heliostats, which results in lower energy efficiency, greater consumption and economic and environmental cost.

Likewise, in the classroom without heliostats, it was observed that the average lighting levels increased in the rear rows of the classroom where the distance to the blackboard was greater, not reaching 100 lx in any case. However, with the installation of the heliostatic illuminators, it was possible to improve the natural lighting conditions in the front and central part of the classroom, where a greater influx of students was expected due to the proximity to the teacher and the blackboard and projector screen.

Furthermore, based on the experimental data recorded, regression analysis was carried out that allowed us to estimate the level of average natural lighting inside the classroom at any time with heliostat illuminators based on global and diffuse solar irradiance data registered outside. Equation (1) show the model obtained in which $\bar{I_{N}}$ represents the average illumination inside the classroom, α the solar height, $I_{G}$ the global solar irradiance and $I_{D}$ the diffuse solar irradiance.
(1)$$\bar{I_{N}}=a\cdot \alpha +b\cdot I_{G}+c\cdot I_{D}+d.$$

Since the independent variables used in Equation (1) are available within the definition of a typical meteorological year, this equation allows us to estimate the average illumination for all the hours of the typical meteorological year.

Table 1 show the values obtained for the constants of the model, (a,b,c,d), which presents a high degree of correlation ($R^{2}=0.802$).

From this model, the average levels of lighting inside the classroom with heliostatic illuminators were estimated during the time the classroom was in use for a full year, considering that classes are held in the classroom from 8:00 a.m. to 8:00 p.m. from Monday to Friday, except for holiday periods and holidays. Figure 18 show the histogram of the estimated average natural lighting levels inside the classroom. It was observed that 37.21% of the time, the level of average natural lighting inside the classroom was equal to or greater than the recommended 300 lx, so it would not be necessary to resort to the use of the complementary artificial lighting system. Moreover, during 62.79% of the remaining time, despite the fact that it would be necessary to use the regulated artificial lighting system, 100% of its power will not be required, which will mean energy savings.

Assuming that the electrical power required (*P*) to supplement the mean light level from value *I* lower than a reference illuminance value (*I_r_*) can be expressed as Equation (2).
(2)$$P=k\left(I_{r}-I\right)$$
where *k* is a proportionality constant.

Considering *p_I_* as the probability of occurrence (value of the histograms) of value *I*, it can be established that the energy required during a month (*E_m_*) to supplement the light levels, when these are deficient, up to a value *I_r_* (300 lx in this work) is provided by Equation (3).
(3)$$E_{m}=\underset{I<I_{r}}{\sum }N\;p_{I}\;P.$$

Equation (3) allows us to establish a proportionality between the energy consumed and the sum, which is evaluable with the information contained in the histograms and the total operating hours (*N*). Therefore, the calculations of these summations allow us to establish the estimated savings in percentage terms.

Specifically, a saving of 64.21% in electricity consumption was estimated, using the lighting level recorded by a luxmeter inside the actual classroom (441 lx) as a reference. Given the high variability of solar irradiance and, consequently, natural lighting, this energy saving was not constant throughout the year. Thus, as an example, Figure 19 show the histograms of the average lighting levels inside the classroom with heliostatic illuminators in the months of February, April and June. It is observed how the frequency of lighting levels below 150 lx decreases in the month of June. Finally, Table 2 show the estimated energy savings for each month of the year (except the month of August because it is not a school month) as a result of the installation of the heliostatic illuminators. As expected, this saving was greater in the months of May, June and July, when the levels of solar irradiance in Cordoba (Spain) are higher. Despite this, in all months, the saving was greater than 50%.

## 4. Conclusions

One of the solutions for the fight against climate change involves the need to improve the energy efficiency of buildings. In this regard, the use of solar resources as a source of direct and indirect energy is essential. As far as its direct use is concerned, the proper use of natural lighting can improve the energy efficiency of buildings. In this line, this paper analysed the possible use of heliostats as devices to improve the levels of natural lighting inside buildings. Specifically, the study focused on the situation of the Leonardo da Vinci building on the Rabanales University Campus of the University of Córdoba (Spain). Despite the high number of hours of sunshine recorded in Córdoba, the design of the building hinders access to the sun inside the classroom. As a consequence, the levels of natural lighting inside the classroom are insufficient for the function of teaching, making it essential to resort to artificial lighting systems at all times.

In these circumstances, in the present study, two scale models of the same classroom of the aforementioned building were developed, and on one of them, an illuminator system based on the heliostat prototype developed by Torres-Roldan et al. [22] was installed. Likewise, the lighting levels inside both models were monitored, and two regulated artificial lighting systems were developed to complement the levels of natural lighting received inside each of them. Both classroom scale models were placed outdoors in a similar orientation to the actual classroom and exposed to sunlight under conditions identical and similar to those in the actual classroom. The qualitative and quantitative comparative analysis of the levels of natural lighting inside both models showed that the levels of natural lighting improve with the installation of the heliostatic illuminators. In fact, it was verified that, while in the classroom without heliostats, the 300 lx recommended for the development of the function of teaching was never reached. In the case of the classroom with heliostats, this level was reached 37.21% of the time during teaching hours in one year. In this way, during that time, it would not be necessary to use artificial lighting in the classroom with heliostats, while in the classroom without heliostats, it would be. In addition, the artificial lighting needs for the remaining 62.79% of the time would be lower in the classroom with heliostats than without heliostats. In this regard, an energy saving of 64.21% is estimated compared to the current use of artificial lighting in the real classroom. To quantify the effects of scale or differences on optical properties, there remains a need to further increase and intensify the measurement periods used in the scale models; therefore, the research line based on the generation of the scale twin should be further developed. In this way, it was verified that the use of heliostatic illuminators can be a viable measure to improve energy efficiency in buildings and contribute to the fight against climate change. Subsequent and future research work, aimed at achieving the physical twin of the real space, will comparatively study the importance of natural lighting effects, such as visual comfort, existence and control of glare and the behaviour of light diffusers in the ceiling. For this purpose, the geometric and optical characteristics of components of the system (skylight, mullions, glass including effects such as dirt) will be assessed. The achievement of the scale model will allow us to spatially characterize the light distribution in the interior, to adequately divide the daylighting system and even optimise the number of heliostats required.

## Figures and Tables

**Figure 1 sensors-22-05996-f001:**
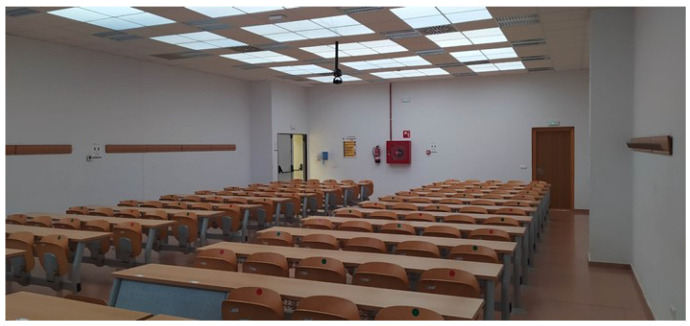
Interior of a classroom in the Leonardo da Vinci building on the Rabanales University Campus of the University of Córdoba (Spain).

**Figure 2 sensors-22-05996-f002:**
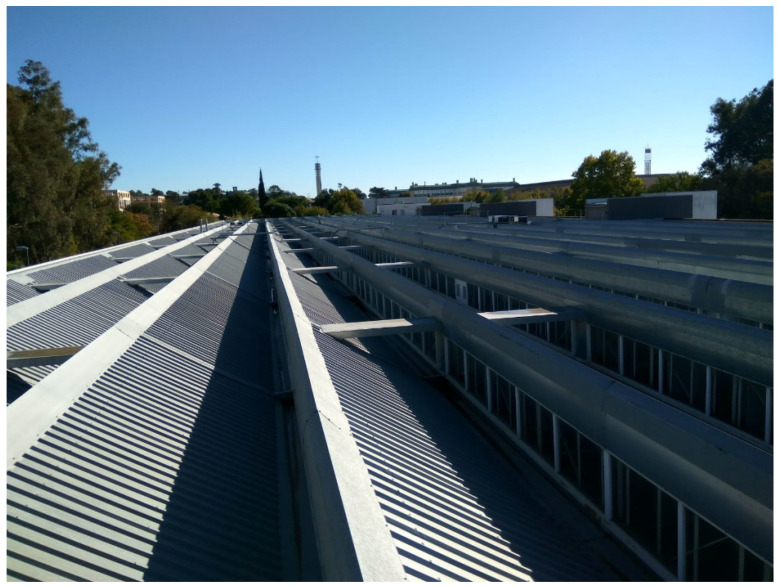
Photograph of the roof of the Leonardo da Vinci building of the Rabanales University Campus of the University of Córdoba (Spain).

**Figure 3 sensors-22-05996-f003:**
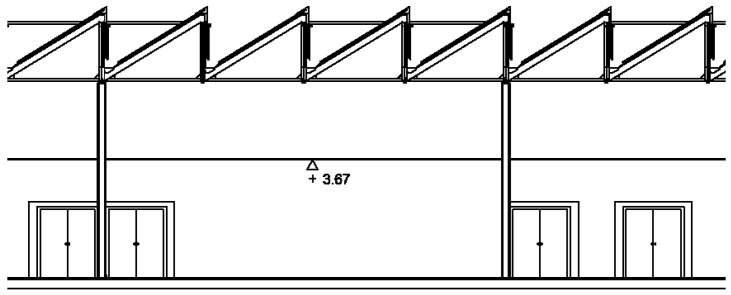
Plan of the cross-section of the Leonardo da Vinci building of the Rabanales University Campus of the University of Córdoba (Spain).

**Figure 4 sensors-22-05996-f004:**
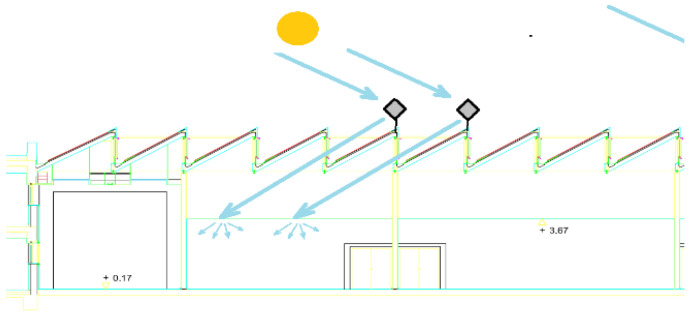
Diagram of installation of heliostatic illuminators on the roof of the Leonardo da Vinci building of the Rabanales University Campus of the University of Córdoba (Spain).

**Figure 5 sensors-22-05996-f005:**
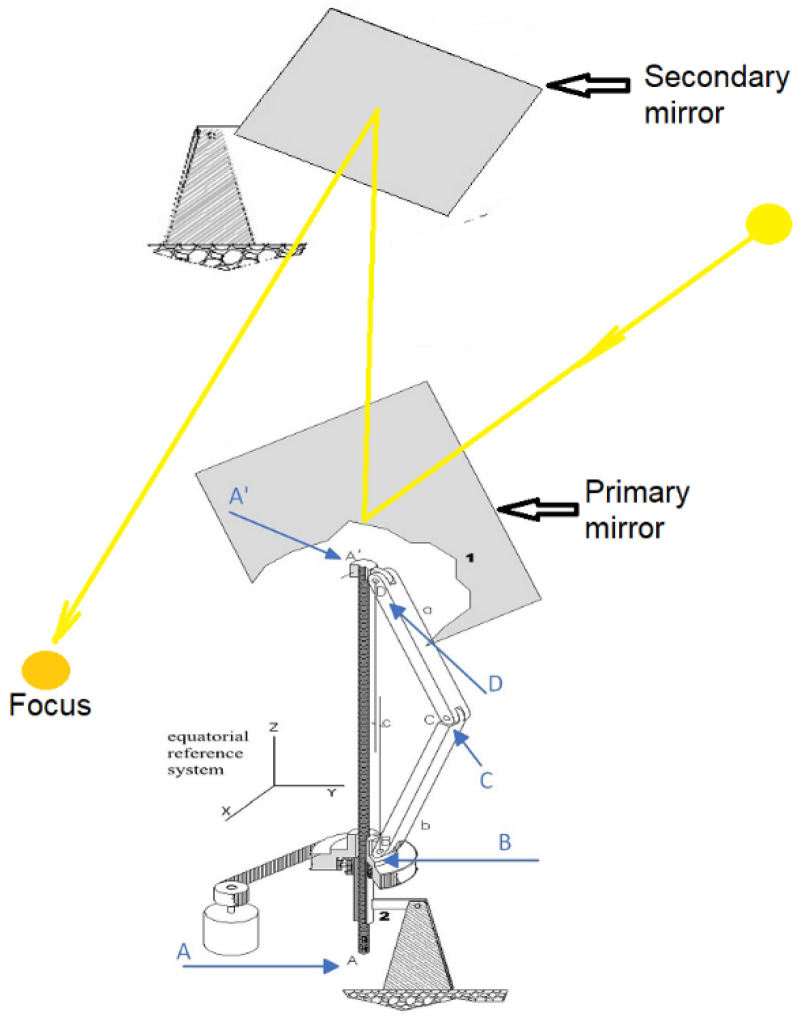
Polar heliostat prototype used for the proposed daylighting system.

**Figure 6 sensors-22-05996-f006:**
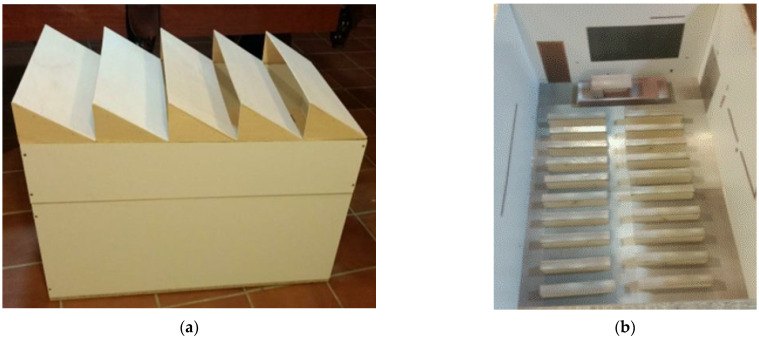
Scale model of the classroom of the Leonardo da Vinci building of the Rabanales University Campus of the University of Córdoba (Spain): (**a**) exterior; (**b**) inside.

**Figure 7 sensors-22-05996-f007:**
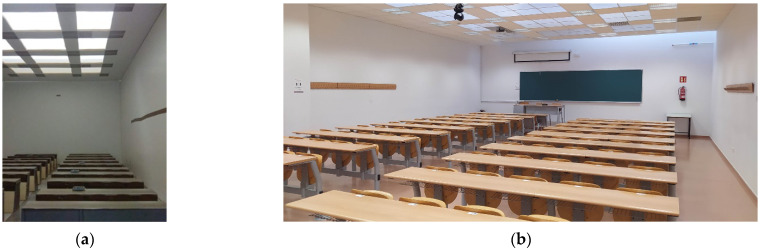
Comparison between the interior of the: (**a**) developed model; (**b**) and the real environment of the chosen classroom of the Leonardo da Vinci building of the Campus of Rabanales at the University of Córdoba (Spain).

**Figure 8 sensors-22-05996-f008:**
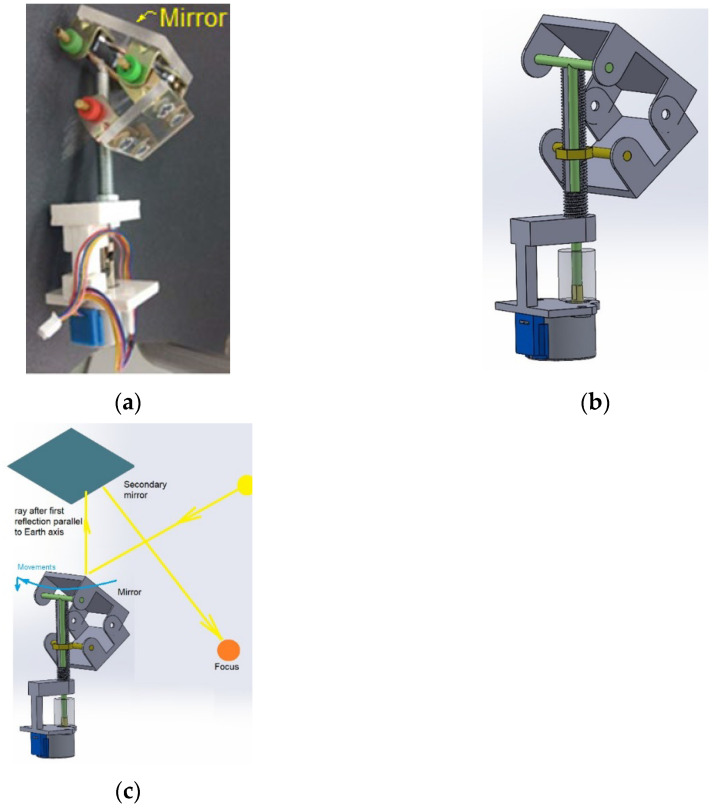
Scale model of the single axis polar heliostats proposed by authors based on Torres et al. (2015). (**a**) Heliostat mechanism used. (**b**) Sectioning of the mechanism used. (**c**) Overall operating principle of the heliostat (blue arrows represent the elementary movements of the primary mirror).

**Figure 9 sensors-22-05996-f009:**
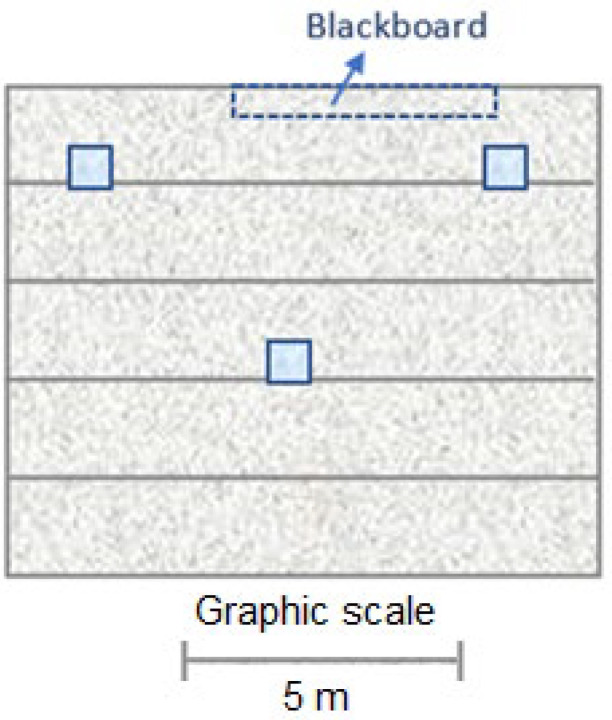
Location plan of the distribution of the heliostatic illuminators in the scale model of the classroom of the Leonardo da Vinci building of the Rabanales University Campus of the University of Córdoba (Spain).

**Figure 10 sensors-22-05996-f010:**
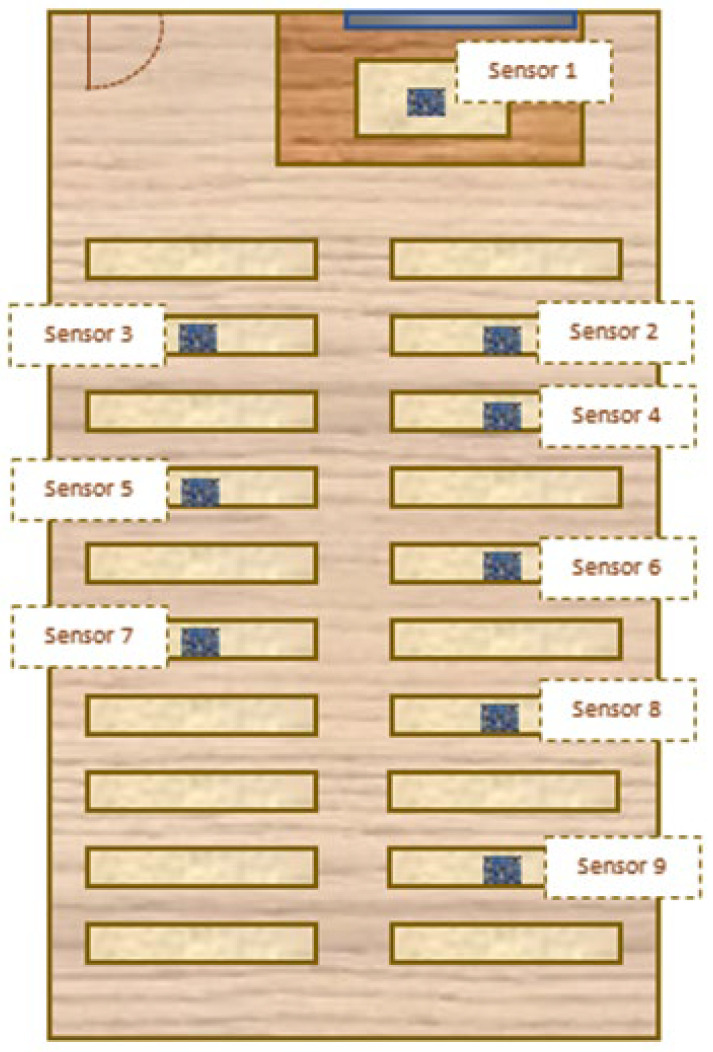
Location scheme of TSL2561 sensors in the scale model of the classroom of the Leonardo da Vinci building of the Rabanales University Campus of the University of Córdoba (Spain).

**Figure 11 sensors-22-05996-f011:**
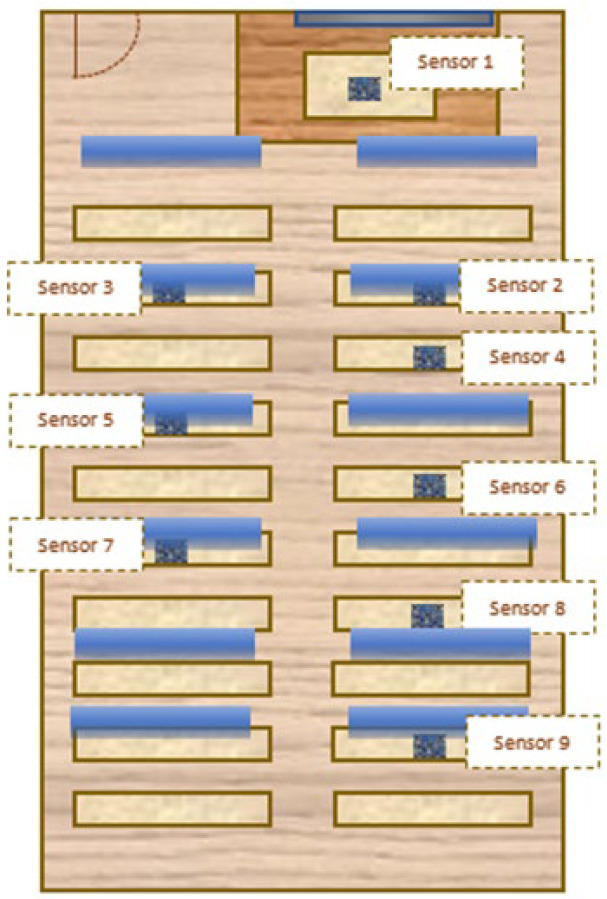
Diagram of the location of the artificial lighting system LEDs in the scale model of the classroom of the Leonardo da Vinci building of the Rabanales University Campus of the University of Córdoba (Spain).

**Figure 12 sensors-22-05996-f012:**
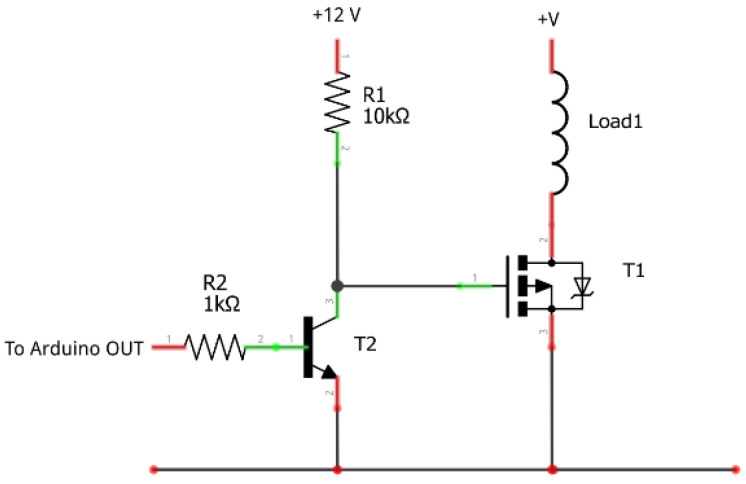
Connection diagram for the intensity regulator of the artificial lighting system based on LED technology.

**Figure 13 sensors-22-05996-f013:**
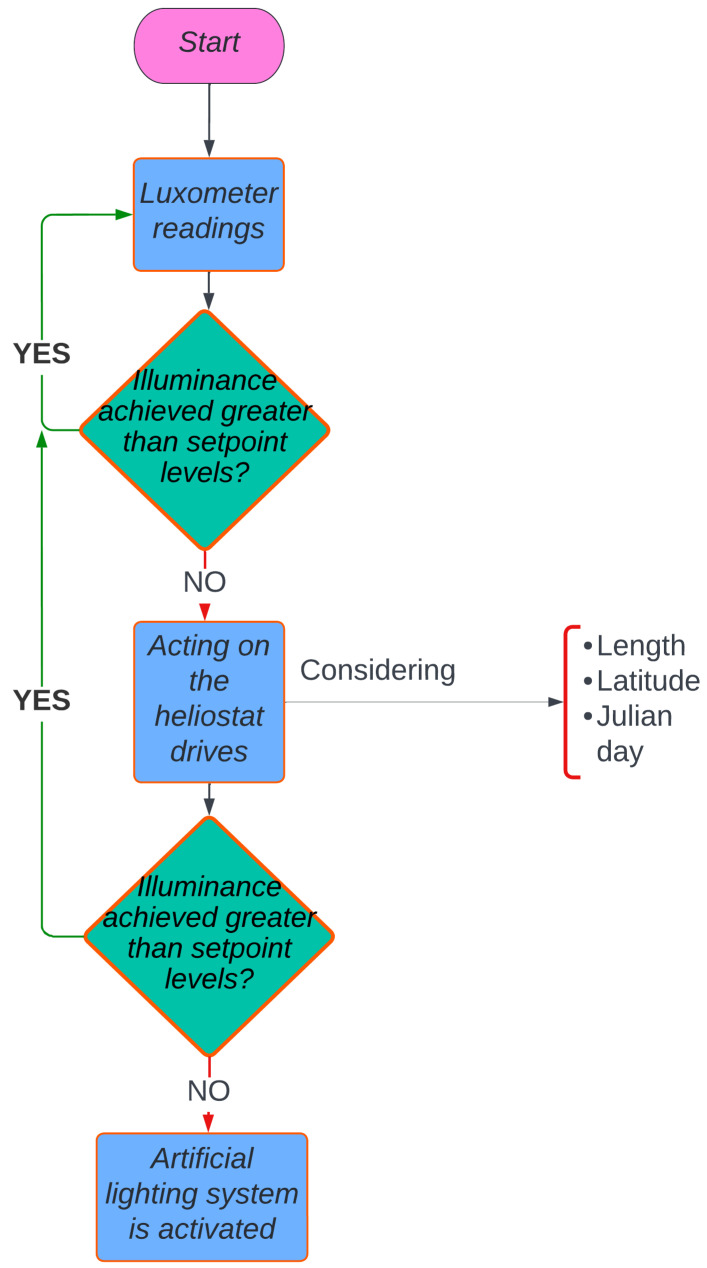
Diagram of the management and programming of the monitoring system.

**Figure 14 sensors-22-05996-f014:**
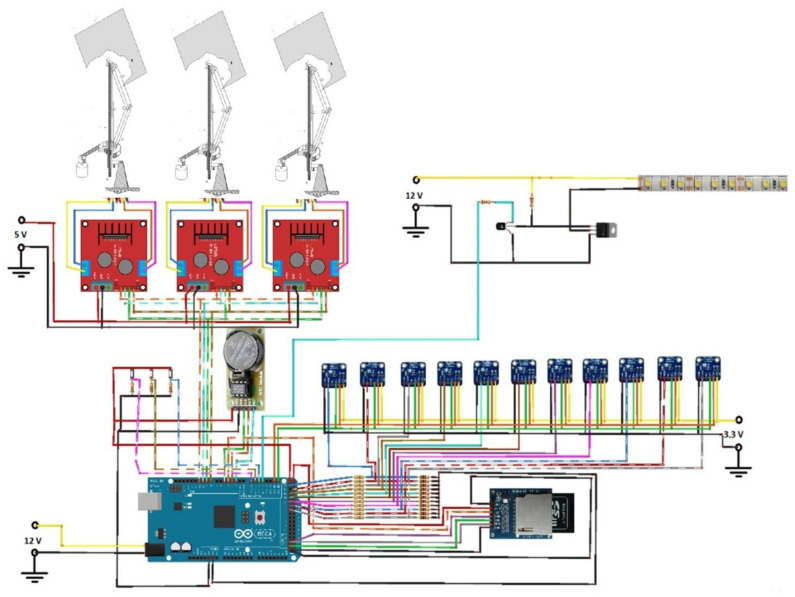
Diagram of connection of the different components to the Arduino Mega 2560 board in charge of the control of the system.

**Figure 15 sensors-22-05996-f015:**
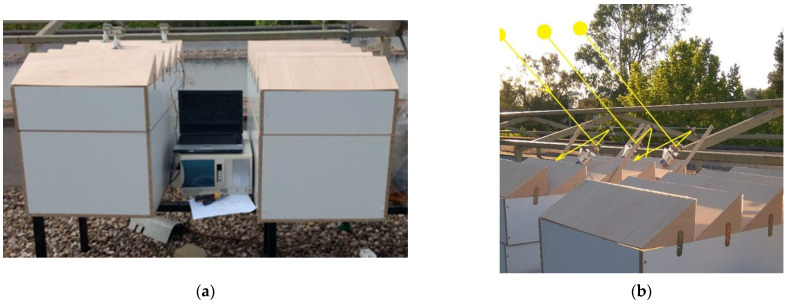
Scale models place outdoors. (**a**) Scale model arrangement and data acquisition system (**b**) Solar rays’ trajectory.

**Figure 16 sensors-22-05996-f016:**
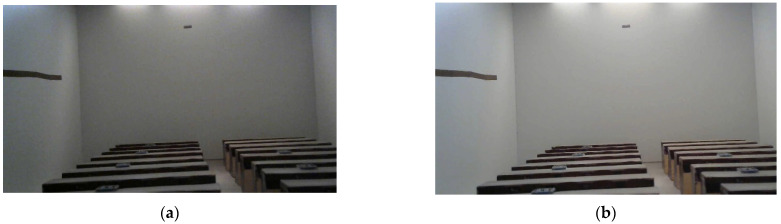
Qualitative comparative analysis of daylighting levels: (**a**) inside the scale model without heliostat illuminators; (**b**) the one with heliostat illuminators.

**Figure 17 sensors-22-05996-f017:**
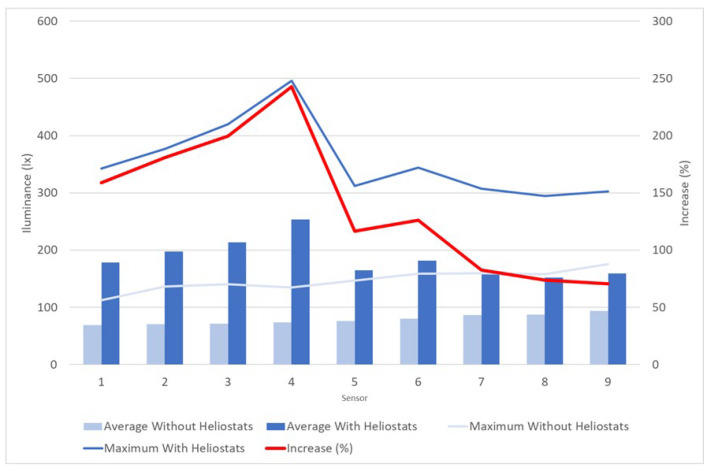
Quantitative comparative analysis of daylighting levels inside the scale model without heliostat illuminators and the one with heliostat illuminators.

**Figure 18 sensors-22-05996-f018:**
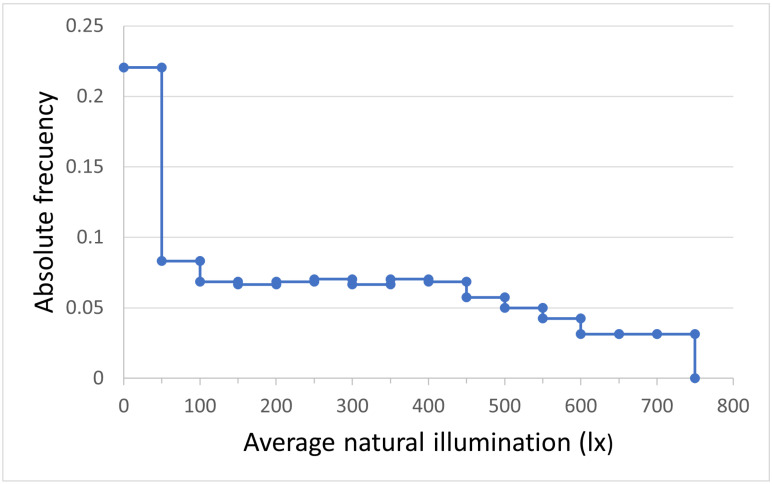
Histogram of average lighting levels inside the classroom with heliostatic illuminators during their time of use for teaching purposes in a full year.

**Figure 19 sensors-22-05996-f019:**
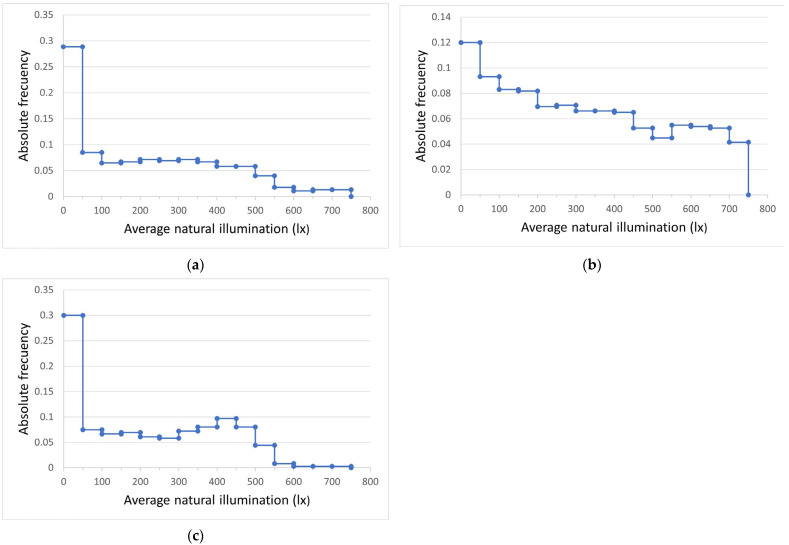
Histogram of average lighting levels inside the classroom with heliostatic illuminators during their time of use for teaching purposes in the months of (**a**) February; (**b**) April; and (**c**) June.

**Table 1 sensors-22-05996-t001:** Values of the constants of the regression model for the average lighting levels inside the classroom with heliostatic illuminators.

Coefficient	Value	Units
*a*	0.641	lx/deg
*b*	0.262	lx·m^2^/W
*c*	0.307	lx·m^2^/W
*d*	0.181	lx

**Table 2 sensors-22-05996-t002:** Savings in estimated monthly energy consumption in the classroom as a result of the installation of heliostatic illuminators.

Month	Saving (%)
January	56.42
February	60.83
March	64.13
April	66.61
May	73.77
June	76.40
July	76.54
September	67.54
October	66.61
November	60.43
December	52.64
January	56.42

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
