# Peer review of "Use of Polar Heliostats to Improve Levels of Natural Lighting inside Buildings with Little Access to Sunlight"

_sensors, 2022, doi:10.3390/s22165996_

Round 1

Reviewer 1 Report

The authors have performed a very interesting study.  My comments on this work are provided below.

First, it is unclear that this paper is appropriate for a journal entitled “Sensors”, since this paper really has very little to do with sensors.  The heliostats do not appear to be controlled by sensors – they can simply be controlled by a time clock, given the site latitude, longitude, and desired direction for the reflected daylight.  The sensors that are applied in the scale models are applied on the work plane, which would not generally be how these would be applied in a real space, and in the study, they appear to simply be recording the daylight work plane illuminance that is provided.  A lighting control system is briefly described, with no details on how work plane sensor readings may impact the operation of the heliostats, if at all.  With the limited details and overall study setup that is outlined in the paper, it is unclear how this system presents new information on the application of sensors in a novel daylighting system.  The sensors appear to control the electric lighting system in response to daylight, which has been common practice for decades.  I do not see any mention that sensors are use to control the heliostats.

Additional comments related primarily to the daylight study conducted are provided below.

1.       It is not stated how long, or when, readings were taken in the models.  Data from these readings were used to compute coefficients for the estimation of interior daylight, but it is unclear if these data cover the entire year or just a portion of it.  If only a portion, then how does the model address differences in the mirror’s changing projected area relative to the solar position that will impact the total lumens that can be redirected?

2.       It is unclear if the author’s heliostat redirecting system involves one or two reflections.  The referenced system by Torres-Roldan et al. involves two mirrors, but the photos of the author’s system only appear to show one.  A better graphic or photograph with arrows showing the sunbeam travel path in the authors’ setup is needed.  Figure 9 has heliostats in multiple different orientations, which is confusing, while Figure 8 does not appear to show a mirrored surface.

3.       The heliostat designed by the authors appears to have significant self-shadowing of the reflected rays by the motor section of the device, if sunlight rays are redirected along the axis of the device by a single mirror.  This would result in a significant reduction in delivery efficiency.

4.       The paper addresses performance only in terms of work plane illuminance.  However, visual comfort and system performance must also consider the appearance and luminance of the lighting system as viewed by the occupants.  Direct reflection of the sun’s rays onto flat lenses at the ceiling plane will create very bright areas on these lenses that are likely to impact both aesthetics and visual comfort.  Photographs in the paper do not show the ceiling and its appearance in the scale models.  Figure 7a, however, does show the presence of what is likely a very bright patch of ceiling in a reflection on the teacher’s desk, confirming the likely potential for glare from high luminances on the underside of the lenses where the sunlight is transmitted into the room.  Further system refinements are necessary to develop a lighting system that is likely to be acceptable in such a space.  The authors also fail to note the material that was applied for the lenses in the scale model and how its optical features relate to those of the lenses used in the real space.

5.       The clerestory opening in the scale model appears to be significantly larger than the window opening in the real space.  All geometry should be appropriately scaled when conducting model studies, with identical reflectances and transmittances applied to all surfaces.  It also appears that no mullions have been applied in the scale models, and no mention of the glazing properties have been provided.  The measured illuminances readings are therefore likely to be significantly higher in the scale model than in the real space.

6.       The stripes for the electric lighting system in Figure 11 are blue, not red as indicated in the text.  Is the row of LEDs at the bottom of this figure intended to be closer to the adjacent row than the spacing utilized elsewhere in the room?  It appears that all are controlled together, which isn’t very logical given the non-uniform layout of the heliostats and the resulting non-uniform distribution of illuminance within the room.  Computing energy savings based on average illuminance ignores that fact that the rear of the room receives less daylight.  More heliostats would be needed to provide a more uniform daylight distribution within the space for such an assumption to be valid.

7.       What is involved in the “YES” path of the control system in Figure 13?

8.       Figure 9a should be drawn to scale.  What is the role of the large triangles that can be seen behind the heliostats in Figure 9b?

This paper could be improved with a better understanding of how the authors’ system operates, since it appears to be configured differently than the one referenced in the paper.  In addition, numerous other details related to the study are missing, and certain aspects of the model and simulations limit the extension of the measured performance to what might occur in a real space. The question of whether this paper contains anything new related to sensors is one that must be addressed by the editor.

Reviewer 2 Report

Thanks for this interesting paper. 

I have few questions, remarks:

Figure 5: the picture is not easy to understand. The letters are not readable. 

Line 200: you have used certain materials (stickers e.g.) with certain optical properties (reflectance, absorption), is this representative for the room? This might have consequences for the amount of light measured from the heliostate system e.g. Can you comment on this?

Related to this: what is the impact of the diffuser in the roof? 

Figure 8: Can you denote the different elements in the figure? The construction of the defoldable mirror is not clear to me.

Figure 12: not readable to me. What do you want to explain.

Line 301: in the system without heliostat the are also no artificial luminaires, correct?

Line 312-324: Are the quoted numbers valid if the reflectance e.g. (see above is not correct)?

Line 366: How did you come to 64,21% (very accurate also, neded)? I cannot do this calculation based on the data presented and I think that should not be the case as this is most important claim.

Round 2

Reviewer 1 Report

A number of my comments were addressed, which improved the paper and will help the reader understand the daylighting system.  Still, there are still a few areas related to the figures where clarity can be improved, as well as details on the control system that were not addressed.

Figure 5, and the text in the body of the paper related to it, need to address the fact that this system requires two mirrors, the first of which directs sunlight along the polar axis direction to a second fixed mirror that redirects light in a desired direction.  Figure 8 appears to be satisfactory for this purpose (and therefore I propose eliminating Figure 5).  In addition, in Figure 8c, I suggest adding arrows and text showing that the lower mirror can both rotate and change in slope to redirect the sun’s rays along the polar axis direction.  The figure title should also be changed to indicate that this is your design of the system developed by Torres. Finally, I suggest adding somewhere that your second mirror is configured to redirect the sunbeam at an angle relative to the clerestory glazing to avoid striking the first mirror that lies below it (this is evident in the new Figure 15, which is very helpful for understanding the system).

The authors should also consider modifying Figure 9, or its caption, since it does not include the second mirror and has the first mirror on the three heliostats each at different orientations.  This is likely to confuse the reader.  Figure 15 may be sufficient, but if Figure 9 is retained, the configuration of the heliostats should be adjusted, or the caption should clarify that the second mirrors are not included.

The text added for the “YES” path in figure 13 still does not clearly describe the connection, in terms of control, between the work plane sensors and the heliostat drives. Does the heliostat ever not follow the sun, or adjust its orientation based on the sensor readings?  Also, how does the electric lighting control system feedback to the operation of the heliostat drives (the arrow between the bottom box and the heliostat box above)?  If there is a control scenario whereby the heliostat drives do not follow the sun, this should be included in the paper.  The paper still provides no information regarding how the work plane sensors do anything besides control the electric light output level.  Specific details are required in the text to explain any feedback from these sensors to the control of the drives.  The authors’ reply to comments on the first draft state “The sensors used in the work have the role of activating the operation of the designed heliostat” but it is unclear what is being done, since the heliostat operation can be completely controlled based on knowledge of the site location and the day/time.
